# Rural Residents' Perceptions, Attitudes, and Environmentally Responsible Behaviors towards Garbage Exchange Supermarkets: An Example from Huangshan City in China

**Song Lu \*, Zehui Zhou and Yingfan Lu**

College of Environmental and Geographical Sciences, Shanghai Normal University, Shanghai 200233, China;
zzh1312759@163.com (Z.Z.); lyfts1229@163.com (Y.L.)
\* Correspondence: ahlusong@shnu.edu.cn

**Abstract:** Rural residents not only produce rural garbage and participate in its disposal, but are also beneficiaries of a beautiful rural environment. The garbage exchange supermarket (where garbage is exchanged for goods) is a garbage disposal method that is employed in some villages in China. It is of great significance for the improvement of rural living environment and rural residents' awareness of environmental protection. Thus, it is necessary to explore rural residents' perceptions and behavior regarding garbage exchange supermarkets. Based on planned behavior theory and social exchange theory, this paper develops a model of rural residents' perceptions, attitudes, and environmentally responsible behaviors regarding garbage exchange supermarkets. Then, using Huangshan City, China, as a case study, three villages, located in the upper, middle, and lower reaches of the Xin'an River were selected. Using a stratified sampling method, 324 questionnaires were obtained from residents. The developed model was verified by the method of structural equation modeling. Findings are as follows: (1) On the whole, residents have a strong and positive perception of the benefits of garbage exchange supermarkets, with an emphasis on its environmental advantages. (2) Regarding the cost dimension in perception, the focus is spent queuing for exchange on the time and sorting garbage at home. In general, residents are still willing to spend this time going to the supermarket to exchange. (3) Environmentally responsible behavior is divided into two dimensions: compliance and promotion-type environmentally responsible behavior—the former is more apparent among rural residents. (4) Residents' perceptions of benefits positively affect their attitudes and satisfaction towards garbage exchange supermarkets. Cost perception has no significant effect on residents' attitudes but has a negative correlation with satisfaction, satisfaction and attitude have positive correlations with environmentally responsible behavior, and satisfaction also positively affects residents' attitudes.

**Keywords:** garbage exchange supermarket; residents' perceptions and attitudes; environmentally responsible behavior; rural residents; Huangshan City

## 1. Introduction

Urbanization is advancing rapidly all over the world. From the United States and Sweden to sub-Saharan Africa, the rural–urban divide is widening, and rural decline is a global issue [1]. Economies in rural areas (especially in developing countries) are relatively backward, with a lack of rural environmental infrastructure and public service facilities. Rural residents' awareness of environmental protection also lags behind their urban counterparts. Thus, there are serious environmental problems in rural areas. Rural garbage disposal and the resulting "dirty and messy" environment have always been a major problem for the rural ecological environment. In order to strengthen the ecological environment in rural areas and improve the living environment, central governments of various countries and local governments at all levels have adopted a series of policies to explore different models of rural garbage management. As a developed capitalist

country, the United States attaches great importance to the use of market mechanisms to solve government problems [2]. By comparison, the German government is more cautious in the field of garbage management and treats rural garbage through urban–rural integration. The government formulates uniform laws and regulations and then collects, transfers, and implements them in a unified manner [3,4]. Japan takes a legal perspective and has established a complete system to ensure the orderly disposal of rural domestic garbage. Japan also attaches great importance to cultivating children's awareness of environmental protection and teaches domestic garbage classification from an early age [5]. Since the beginning of the 21st century, the Chinese government has stepped up efforts to support rural development and put forward strategies such as the new socialist community construction, developing a beautiful countryside, and the revitalization of rural areas, in order to promote the improvement of the rural living environment and the management of obvious problems in the rural ecological environment [6]. In this context, a rural garbage management model, termed a "garbage exchange supermarket", has emerged in rural China, providing an opportunity to solve the garbage problem in rural areas.

The origin of China's rural garbage exchange supermarkets can be traced back to Shaanxi Province in 2011. In response to the serious problem of local garbage in rural areas, the local government initiated a garbage treatment model of "villagers collect and classify, exchange materials at fixed points and government subsidizes the difference", which was well-received by local residents. Villagers brought recyclable garbage, such as wine bottles, mineral water bottles, and wastepaper to the supermarket in exchange for daily necessities. Thus, the garbage is collected effectively and classified simultaneously while the villagers not only gain benefits but also improve their rural living environment. The establishment of the garbage supermarket has brought considerable improvement to the local environment. Attached to the original village supermarket or canteen, the garbage exchange is an incidental function. Garbage exchange supermarkets have been widely piloted in Shaanxi. Since 2015, the concept has been gradually introduced in Quzhou in Zhejiang and Suzhou in Jiangsu. In other places, such as Huangshan in Anhui, the implementation effect is more significant.

Long before this, rural modernization in western countries developed rapidly, leading to early exposure of the severe problems of garbage management in rural environments and generating much research interest. These studies mainly focus on the infrastructure of rural garbage treatment [7–9], governance behavior [9–12], and governance models [2–5]. Similarly, related research also examines waste treatment from the residents' perspectives [7]. In China, research on waste management mainly deals with explorations of its models [13,14] and the legal system [15–18]. In recent years, research has also begun to explore factors which influence residents' participation in waste management. For example, Xie et al. (2020) extended the theory of planned behavior (TPB) to explore the willingness of rural residents to participate in rural governance [19]. Based on data from 327 rural residents, the TPB was combined with the theory of normative activation to analyze the garbage classification behavior of rural residents [20]. Qianqian Xu used the DEA model to measure the efficiency of rural residents' participation in domestic waste classification and suggested its importance in improving the efficiency of waste management [21]. The environmentally responsible behavior of rural residents has also become a focus of research. Oluyinka and Ojedokun proposed that environmentally responsible behavior is a habitual feature or conscious behavior of rural residents, which can prevent destructive behavior in daily activities (such as intentional or unintentional littering) [22]. Foguesatto and others considered the sustainable behavior of rural residents, referring to practices that contribute to economic, social, and environmental aspects, such as the use of green fertilizers and other environmentally responsible behaviors [23]. On the basis of typical planned behavior, norm-activation, and value-belief norm theories, the factors influencing rural residents' environmental responsibility behavior have been examined in different situations [24–26]

to regulate such practices and improve environmental problems. These findings provide practical implications.

Rural residents are not only the producers of garbage and participants in garbage management but are also the beneficiaries of the construction of beautiful villages. Therefore, in the context of garbage management, it is necessary to study the perception and behavior of residents from the residents' perspectives. As a product of the combination of the Xin'an River Basin's ecological compensation system pilot and renovation of human settlements, Huangshan City initiated a rural garbage exchange supermarket in 2016 and upgraded it into an ecological beauty (eco-beauty) supermarket in 2018. By 2021, a total of 345 supermarkets had been built [27]. As a special rectification measure, garbage exchange supermarkets play a key role in rural environmental governance in Huangshan City and provide a solution for rural garbage disposal. Taking the Huangshan garbage exchange supermarket as the starting point, this study analyzes the perceptions and attitudes of rural residents towards the garbage exchange supermarket and its relationship to environmentally responsible behavior. The environmental awareness and environmental behavior of rural residents are also explored to provide a reference for the improvement of the rural living environment and rural revitalization.

## 2. Model Building

Rational behavior theory is one of the basic theories of cognitive behavior in social psychology, resulting in its considerable theoretical influence and application. Its main point is: "Individual behavioral attitudes and subjective norms will lead to behavioral intentions, and the behavioral intentions of individuals with willpower will directly affect actual behaviors". However, in many cases, behavior is not entirely controlled by the individual's will. The implementation of specific individual behaviors often requires resources such as time, energy, money, skills, and corresponding facilities and equipment; that is, the implementation of specific behaviors is often affected by many non-volitional factors. On this basis, Ajzen added a non-volitional factor to perceive behavior control based on rational behavior theory, transforming it to form the theory of planned behavior (TPB) [28]. Numerous empirical studies have verified the effectiveness of TPB in explaining and predicting individual behavior. This theory is also widely used in the field of environmentally responsible behavior. For example, regarding relevant decision-making such as in tourists' travel period [29], rural residents' agricultural production pollution control [30], rural ecological environment supervision [31], and rural residents' participation in garbage classification [32], the TPB model can provide effective explanations. In addition, social exchange theory, as one of the widely used theoretical frameworks, can be used to explain the motivation of residents to participate in certain activities [33]. As rational people, residents who participate in an activity evaluate whether they can generate benefits and pay costs and make a trade-off between the two. Whether the activity is satisfactory or not can affect the residents' participation behavior; thus, evaluating these benefits and costs has become an important factor affecting residents' participation behavior. Based on the combination of TPB and social exchange theory, this study selects residents' perceptions, attitudes, satisfaction, and environmental responsibility behavior of garbage exchange supermarket as variables, puts forward relevant research hypotheses, and constructs a structural model of rural residents' perceptions of garbage exchange supermarket and environmentally responsible behavior.

### 2.1. Perception, Attitude, and Environmentally Responsible Behavior

Social exchange theory studies social behavior from the perspective of the input–output ratio of economics. As rational economic people, rural residents' environmentally responsible behavior is related to the benefits they receive and the costs they need to pay. Therefore, when studying the environmentally responsible behavior of rural residents, income perception and costs have a significant impact. Khamfea et al. studied local residents' perceptions, attitudes, and participation towards Laos' national protected areas. Benefit

perception has a significant positive impact on protective attitudes and behaviors [34]. Wei Duan used data from Shaanxi, Hunan, and Jiangxi to analyze the benefit and cost perceptions of rural residents around protected areas. Most residents have a positive attitude and support the establishment of national protected areas [35]. Anqi Chen et al. evaluated residents' garbage exchange behavior and concluded that the benefits of garbage exchange supermarkets outweigh the costs. At the same time, the management model of garbage exchange supermarkets also affects residents' attitudes toward supermarkets, which in turn affects their environmentally responsible behaviors [36]. The participation of rural residents in garbage exchange also depends on the benefits they gain and costs they pay, which affect their attitudes towards garbage exchange supermarkets and related environmentally responsible behaviors. Therefore, the relationships between benefit perception and attitude, benefit perception and behavior, cost perception and attitude, and cost perception and behavior have been mainly confirmed in the literature. Accordingly, this study proposes the following hypotheses:

**Hypothesis 1 (H1).** *Rural residents' perceptions of the benefits of garbage exchange supermarkets are positively correlated with their attitudes towards protection.*

**Hypothesis 2 (H2).** *Rural residents' perceptions of the cost of garbage exchange supermarkets are negatively correlated with protection attitudes.*

Rural residents' behavior is not only economic but also social. Thus, social psychologists have developed models to predict and explain the environmental perception and behavior of rural residents in various scenarios. In social psychology, TPB is one of the most important theories [37]. Its three paths, which connect attitude and behavior, have been extensively confirmed. In relevant literature on the driving factors of environmental responsibility behavior, the positive impact of attitude on individual environmental behavior has also been widely confirmed [38]. Jiehong Zhou used TPB to explain the connection between the attitude and the vegetable supply behavior of rural residents, verifying the relationship between home and family [39]. Ling Nan et al. also analyzed the attitudes affecting rural residents' farmland protection behavior and supplemented the indicators of attitude variables [40] that affect their environmentally responsible behavior. Accordingly, this study proposes the following hypothesis:

**Hypothesis 3 (H3).** *Residents' attitudes towards garbage exchange supermarkets are positively correlated with their environmentally responsible behaviors.*

### 2.2. Perception and Satisfaction

Satisfaction refers to obtaining a kind of psychological satisfaction in an experience and is also a kind of evaluation [41]. The relationship between perception and satisfaction has been confirmed in many fields. In public services, customers are the main service evaluators and the value they perceive has become an important indicator of satisfaction. Yan Jiang confirmed that customer service perception has a direct impact on satisfaction [42]. By building a model, Junyi Zheng explored the internal mechanism of the public's perception of satisfaction to behavioral intentions [43]. Jing Li constructed an authenticity perception–satisfaction model in the context of tourism and proposed that authenticity perception has a significant positive effect on satisfaction [44]. Based on data from 556 farmers in Hebei Province, Liu Qingqiang verified that rural residents' perceived value of their new dwellings directly affects their satisfaction with them [45]. When residents participate in activities, the trade-off between the perceived benefits and the costs directly affects their satisfaction. Therefore, based on this background, from the perspective of benefit and cost perceptions, this study proposes that residents' perceptions of garbage exchange supermarkets can affect their satisfaction. Accordingly, this study proposes the following hypotheses:

**Hypothesis 4 (H4).** *Rural residents' perceptions of the benefits of garbage exchange supermarkets are positively correlated with satisfaction.*

**Hypothesis 5 (H5).** *Rural residents' perceptions of garbage exchange supermarket cost are negatively correlated with satisfaction.*

*2.3. Satisfaction, Attitude, and Environmentally Responsible Behavior*

Environmentally responsible behavior emphasizes the active protection of the environment and its relationship with satisfaction is extensively confirmed. For example, Tsung Hung Lee found that tourist satisfaction has a significant positive effect on their environmentally responsible behaviors [46]. Xiaonan Wang explored the influencing factors of garbage classification behavior in Shanghai and proposed the positive effect of satisfaction [47]. Zhi Wang also demonstrated the relationship between satisfaction and behavior by investigating the participation and influencing factors of residents in Tongling City on garbage sorting [48]. Moreover, Hu Chen, Qing Mei, and other scholars found that satisfaction not only has an effect on environmentally responsible behavior, but also has different degrees of influence on promoting other factors of environmentally responsible behavior [49]. In general, the higher the degree of satisfaction, the more positive the attitude presented. For example, Ben Ma found that the more satisfied rural residents are with protected areas, the more positive their attitude toward their continued protection [50]. In the context of the implementation of garbage exchange supermarkets, satisfaction is, to a certain extent, residents' feelings after participating in the exchange. Satisfaction not only affects such participation but also the residents' environmentally responsible behaviors and attitudes. Accordingly, this study proposes the following hypotheses:

**Hypothesis 6 (H6).** *Residents' satisfaction with garbage exchange supermarkets is positively correlated with attitudes.*

**Hypothesis 7 (H7).** *Residents' satisfaction with garbage exchange supermarkets is positively correlated with environmentally responsible behavior.*

Following the proposed hypotheses, a model was developed to illustrate the relationship between perception, attitude, and environmentally responsible behavior of garbage exchange supermarket (see Figure 1).

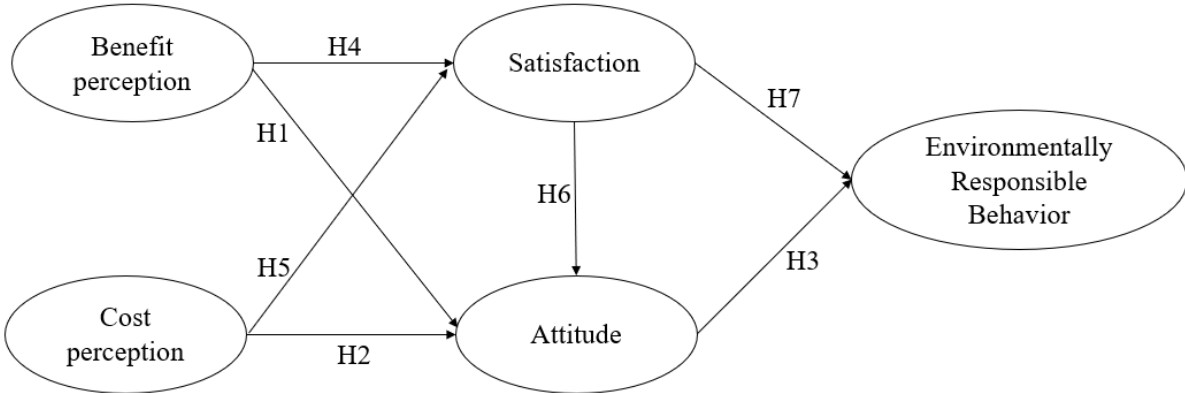

**Figure 1.** Relationship model between rural residents' perceptions and environmentally responsible behavior.

**3. Study Design**

*3.1. Case Outline*

Huangshan City is located at the junction of the three provinces of Anhui, Zhejiang, and Jiangxi, with a total area of 9807 km². The city has jurisdiction over Tunxi District,

Huangshan District, Huizhou District and She County, Xiuning County, Yi County, and Qimen County. The terrain is dominated by mountains, which account for 80% of the land area, has rich biological resources and a superior natural environment. The forest coverage rate is 82.9%, known as the national forest city. The area has beautiful landscapes, a rich history, customs, and traditions. It is a famous tourist destination in China. The Xin'an River originates from Liugujian Mountain in Xiuning County, Huangshan City, Anhui Province, and flows eastward into western Zhejiang Province. The total length of the mainstream is 359 km, of which the section in Anhui Province is 242.3 km long, and the total basin area is about 11,452.5 km$^2$, mainly located in Huangshan City, Anhui Province (accounting for nearly 60% of the area). In order to protect downstream water quality, Anhui and Zhejiang provinces have (since 2012) launched pilot projects involving a cross-provincial ecological compensation system in the Xin'an River Basin. So far, three rounds of nine-year pilot projects have been completed, creating a national precedent for ecological compensation in the inter-provincial river basin, and is a typical case of the ecological civilization construction being implemented by the Chinese government.

Huangshan City is the main organizer of ecological protection in the upper reaches of the Xin'an River Basin. The city has explored and formed the "Xin'an River model" of cross-provincial river basin ecological compensation, and its pilot has achieved productive outcomes. One important achievement is the innovative garbage exchange supermarket (also known as an ecological beauty supermarket). In July 2016, Liukou Town, in Xiuning County, Huangshan City, established the first garbage exchange supermarket. In September 2018, Huangshan City comprehensively promoted the garbage exchange supermarket in the Xin'an River Basin, forming a national protection and innovation approach of "government guidance, market supplemented, public participation and ecological sharing". This model mainly adopts the "barter" method, whereby garbage is exchanged for daily necessities. For example, 10 mineral water bottles can be exchanged for a bag of rice wine or a toothbrush, and 5 old batteries or 60 cigarette boxes can be exchanged for a bag of salt. At the beginning of the establishment of the garbage exchange supermarket, residents are informed by letter, and a notice on bulletin board of the village to publicize waste classification. At the same time, several teams are sent to relevant villages and towns to guide and train residents on waste classification and improve their awareness of and participation in such practices. Residents have come to exchange garbage, which has solved the problem of garbage pollution at the source and improved the ecological environment. Garbage exchange supermarkets have a centralized collection area where garbage is sorted by staff. Recyclable items are sold and the funds are used to subsidize the supermarket's operations. Non-recyclable items are transported to the garbage disposal center. Estimates show that the average monthly purchase amount of each eco-beauty supermarket is 2000 yuan. The income from the sale of waste is 170 yuan per month; thus, the monthly shortfall of funds is approximately 1830 yuan. To solve this problem, the management adopts various methods to supplement funds—government subsidies are the main source. Social subsidies and collective investments are also used as supplementary funding.

After continual practice and improvement, Huangshan City has developed a unique and distinctive "garbage exchange supermarket" model. First, the model has developed rapidly and has a wide coverage. In December 2019, the city had 172 garbage exchange supermarkets, including 61 in Xiuning County, 47 in the Huizhou District, 35 in Shexian County, 16 in Qimen County, 8 in Yixian County, and 5 in the Huangshan District. The locations cover 83 townships and 172 administrative villages/communities (see Figure 2). Second, substantial benefits are apparent and the residents' cooperation is high. The establishment of the garbage exchange supermarket has produced wide-ranging benefits relating to ecology, poverty alleviation, economy, and society in the area. For example, in terms of ecological effects, an average of over 100,000 plastic bottles, more than 10,000 cans, and more than 5000 plastic bags are recycled each year in each eco-beauty supermarket. In addition, a strictly regulated daily operation system and a stable funding channel for replenishment have been formed.

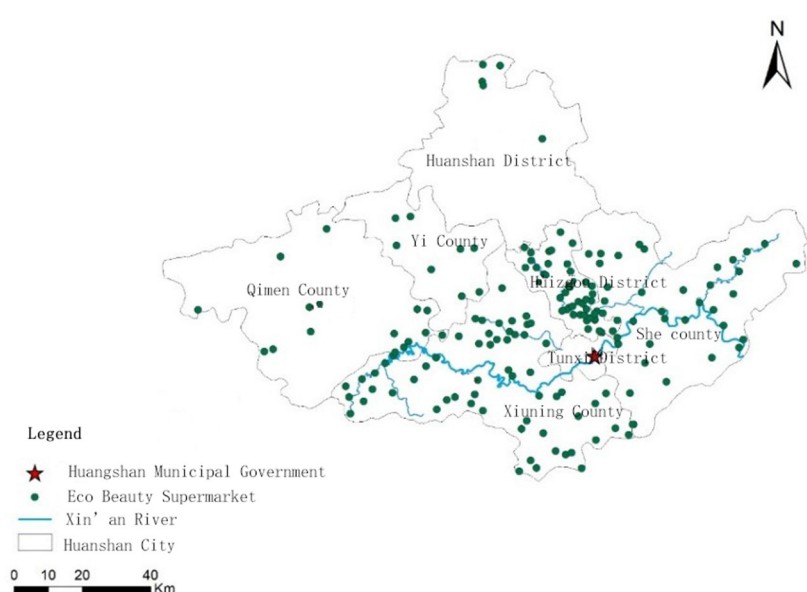

**Figure 2.** Distribution of garbage exchange supermarkets in Huangshan City (end of 2019).

### 3.2. Questionnaire Design

The questionnaire development has three steps, which are described below.

① Design and optimization. Based on domestic and foreign literature on perception, attitude, and behavior, combined with residents' interview data, media data, and characteristics of garbage exchange supermarkets, a measurement scale for residents' perceptions and attitudes is selected and designed. The perception dimension is based on the scale developed by Sirivongs and other scholars [34]. The items are screened and optimized, and perception is divided into two dimensions: benefit and cost perceptions. "Attitude" refers to Adams [8] and Yoon [51], with the help of the research of Kaiser [52] and Sirivongs [34] to determine the division of environmentally responsible behavior. "Attitude and Satisfaction" refer to Chen Anqi's survey on the satisfaction with garbage exchange supermarkets. This current study refers to the above scale and designs the dimensions of attitude and satisfaction [36].

② Pre-investigation. A pre-investigation was carried out on 10 August 2020 and resulted in 38 valid questionnaires. IBM SPSS20.0 software was used to input and analyze the data as well as to make necessary modifications and adjustments. The factor loadings of each measurement index met the requirements, and in line with the preset dimensions, a standard scale with 19 items was finally formed.

③ Formal questionnaires are formed. The questionnaire includes demographic details of the respondents, the characteristics of garbage exchange supermarkets, and the abovementioned 19 measurement indicators, and opinion consultation. The indicators are measured using a 5-point Likert-type scale, where "1" to "5" represent "strongly disagree" to "strongly agree". The consultation section consists of two open questions: Do you have any opinions and suggestions on the construction of rural garbage exchange supermarkets and garbage classification? What are your opinions and suggestions on the improvement of the rural living environment?

### 3.3. Data Collection

The investigation team of six people distributed questionnaires in Huangshan City from 13 to 18 October 2020. The investigators were specially trained, then visited Liukou village, Liukou town, and Xiuning County in the upper reaches of the Xin'an River Basin (the first garbage exchange supermarket in Huangshan City), Xixinan town and Huizhou District in the middle reaches, and to Huangtian village, Huangtian township, and Shexian County in the lower reaches. The questionnaire survey adopts a two-step stratified random

sampling method. First, the target sample size of each district sampling location is determined according to the population ratio and error estimation among the three districts of Huangshan City. Second, the quota sampling method is used. The overall sample is stratified according to household registration, the sample size is determined and the survey objects are selected according to random sampling within the quota. According to the proportion of their resident populations, in Liukou village, Huangtian village of Huangtian township of Shexian County, and Xixinan village of Xixinan town, questionnaires were distributed on a per household basis in Liukou village of Liukou town (123 copies), Huangtian township of Shexian County Tamura (50 copies), and Xixinan village of Xixinan town (177 copies). A total of 350 questionnaires were distributed and 340 were recovered. The exclusion of those with incomplete and apparently random answers yielded 324 valid questionnaires.

## 4. Findings

### 4.1. Basic Information of Respondents

In the sample of rural residents participating in the garbage exchange supermarket exchange (see Table 1), the proportion of women is higher than that of men. Most rural residents are over 60 years old, and the lowest proportion of rural residents is between 20–29 years old. Among the residents, the educational level is mainly primary school and below; the annual income is mainly 5000 yuan and below, with those earning 10,000–15,000 comprising the smallest group. Farming is the main occupation and most rural residents not only engage in agriculture but also go out to work during quiet farming seasons. Almost all respondents were locals, accounting for 99.1% of the sample, and very few were temporary residents.

**Table 1.** Demographic characteristics of the surveyed rural residents.

| Project | | Frequency | Percentage |
|---|---|---|---|
| Sex | Male | 84 | 25.9% |
| | Female | 240 | 74.1% |
| | Under 20 | 0 | 0% |
| Age | 20–29 years old | 9 | 28% |
| | 30–39 years old | 19 | 5.9% |
| | 40–49 years old | 23 | 7.1% |
| | 50–59 years old | 94 | 29% |
| | Over 60 years old | 179 | 55.2% |
| | Elementary school and below | 222 | 68.5% |
| Education | Junior high school | 70 | 21.6% |
| | High school and secondary school | 17 | 5.2% |
| | College | | |
| | Bachelor's degree and above | 7 | 2.2% |
| | 5000 and below | 6 | 1.9% |
| | 5000–10,000 yuan | 225 | 69.1% |
| Average annual income | 10,000–15,000 yuan | 39 | 12% |
| | 20,000 yuan | 27 | 8.3% |
| | Farming | 34 | 10.5% |
| | Business | 174 | 46.3% |
| | Forestry/tea tndustry | 34 | 10.5% |

**Table 1.** *Cont.*

| Project | | Frequency | Percentage |
|---|---|---|---|
| Source of income | Work to earn a living | 88 | 27.2% |
| | Freelance | 53 | 16.4% |
| | Others | 16 | 4.9% |
| | Yes | 18 | 5.6% |
| | No | 321 | 99.1% |
| Local resident | | 3 | 0.9% |

The results of the survey found that almost all residents participate in garbage exchange activities (99.7%), and the coverage rate of villagers is very high. Most of the exchanged garbage comprises daily waste (see Table 2), of which plastic bags are the highest category (85.5%), followed by plastic drinking bottles, cigarette shells, and cigarette butts. For exchange items, residents mainly choose daily necessities, among which salt is the highest (62.3%), followed by detergent (55.6%). Most residents (52.5%) visit garbage exchange supermarkets twice a month, or once a week (16.7%) for those who generate more household waste. Most areas have not implemented a points management system, so the participation rate in those schemes is not high. In the sample, only 24.7% of residents have participation-related points. When asked how they learned about the exchange supermarket, 56.5% were informed by friends and neighbors, 47.8% were informed by "announcements posted in the village", and 19.1% were informed about the supermarket through the distribution of "a letter to residents". These results show that the spread of information in rural areas is mainly by word of mouth. The garbage exchange supermarket also held related activities, such as agricultural product exhibitions and sales. Only 5.9% of the residents said they participated in such related activities. For ecological compensation, most of the residents are unaware of the concept and only 10.3% have heard of, and have a certain understanding of, ecological compensation.

**Table 2.** Characteristics of rural residents participating in garbage exchange.

| Project | Frequency | Percentage | Project | Frequency | Percentage |
|---|---|---|---|---|---|
| **Exchange type** | | | **Redemption frequency** | | |
| Plastic bag | 277 | 85.5% | 1 time/week | 54 | 16.7% |
| Cans | 43 | 13.3% | 2 times/month | 170 | 52.5% |
| Plastic drinking bottle | 231 | 71.3% | 2 times/year | 68 | 21 % |
| Paper drinking bottle | | | 1 time/year | 31 | 9.6% |
| Cigarette case | 62 | 9.1% | Other | 1 | 0.3% |
| Cigarette butt | | | **Participation points system** | | |
| Other | 159 | 49.1% | Not implemented | 199 | 61.4% |
| **Exchange item type** | 129 | 39.8% | Implemented, not participating | 45 | 13.9% |
| Bbutter /soy sauce | 30 | 9.3% | Implemented, participated | 80 | 24.7% |
| Toothbrush | | | **Learned about the supermarket via** | | |
| Toothpaste | | | Friends and neighbors | 183 | 56.5% |
| Dish soap | 103 | 31.8% | Letter to the villagers | 62 | 19.1% |
| Toilet paper | 66 | 20.4% | Village post announcement | 155 | 47.8% |
| Salt | 84 | 25.9% | Other | 8 | 2.5% |
| Chicken essence | 180 | 55.6% | **Understanding of ecological compensation** | | |
| Other | 72 | 22.2% | Know about eco-compensation | | |
| | 202 | 62.3% | Heard of ecological compensation | 34 | 10.5% |
| | 14 | 4.3% | Do not know about ecological compensation | 42 | 13% |
| | 7 | 2.2% | | 248 | 76.5% |

*4.2. Exploratory Factor Analysis*

4.2.1. Rural Residents' Perceptions of Garbage Exchange Supermarkets

The reliability of the scale and each dimension index was analyzed by SPSS software. The results show a good reliability for each dimension. The Cronbach coefficients of benefit perception, cost perception, attitude, satisfaction, and other dimensions are between 0.791–0.861, which are all greater than 0.7, and the overall reliability reaches 0.757, indicating that the data can be further analyzed. The exploratory factor analysis was conducted on the indicators perceived by rural residents. The results show that the KM0 value was 0.779, which was greater than the recommended value of 0.700, and the Sig value of the Bartlett sphericity test was less than 0.001, indicating that the data were suitable for factor analysis. The maximum variance method was used to rotate and extract the indicators with eigenvalues greater than 1 and factor loadings not less than 0.5 as selection criteria to select the index item with the strongest explanatory power. Finally, 16 main indicators were obtained, the eigenvalues were all greater than 1, and the cumulative variance contribution rates were 17.988%, 35.185%, 50.512%, and 65.315%, respectively. After rotation, each factor loading was greater than 0.5 and the maximum was 0.888.

Table 3 shows further analysis of data. Residents' attitude values are the highest, with an average value of 4.37, followed by benefit perception of 4.33. The average satisfaction rate is 4.15, indicating that residents' positive perceptions of these variables are relatively strong. However, residents' perceptions of the cost of garbage exchange supermarkets are weak, with an average value of only 2.27. The standard deviations of the mean values of these variables are between 0.53 and 1.16. Overall, the residents' perceptions and attitudes are relatively consistent.

**Table 3.** Analysis of the overall characteristics of rural residents' perceptions.

| Dimensions and Items | Mean Statistics | Standard Deviation Statistics | Factor Loadings | Variance Contribution Rate |
|---|---|---|---|---|
| Benefit perception | 4.33 | 0.53 | | 17.988% |
| B1 Spam exchange improves local image | 4.27 | 0.623 | 0.715 | |
| B2 Enhance residents' awareness of environmental protection | 4.31 | 0.608 | 0.657 | |
| B3 After the establishment of the supermarket, the waste in the village has been reduced | 4.43 | 0.642 | 0.763 | |
| B4 Establishment of the supermarket contributes to the air protection | 4.30 | 0.628 | 0.803 | |
| B5 Establishment of the supermarket contributes to the water protection | 4.39 | 0.622 | 0.810 | |
| Cost perception | 2.27 | 1.01 | | 35.185% |
| C1 Need to spend time on learning and training related to garbage classification | 2.14 | 1.036 | 0.883 | |
| C2 Takes time to sort waste | 2.35 | 1.121 | 0.871 | |
| C3 You need to wait in line for garbage exchange | 2.40 | 1.161 | 0.888 | |
| Satisfaction | 4.15 | 0.701 | | 50.512% |
| A1 The exchange of garbage in the supermarket is very practical and safe | 4.28 | 0.741 | 0.825 | |
| A2 Garbage exchange supermarkets exchange a wide range of garbage types | 4.06 | 0.804 | 0.785 | |
| A3 Garbage exchange supermarkets is very convenient and affordable | 4.22 | 0.721 | 0.836 | |
| A4 The publicity model and the management of the garbage exchange supermarket are very suitable | 4.02 | 0.851 | 0.814 | |
| Manner | 4.37 | 0.618 | | 65.315% |
| D1 The establishment of the garbage exchange supermarket contributes to the improvement of the environment in the region | 4.24 | 0.742 | 0.753 | |
| D2 I am satisfied with the current management of the garbage exchange supermarket by the supermarket manager | 4.23 | 0.687 | 0.729 | |
| D3 I hope the garbage exchange supermarket will continue to run | 4.43 | 0.745 | 0.800 | |
| D4 I support the establishment of a garbage exchange supermarket | 4.50 | 0.710 | 0.851 | |

4.2.2. Environmentally Responsible Behavior of Rural Residents

Exploratory factor analysis is also carried out on the indicators of rural residents' environmentally responsible behavior. Table 4 shows the results. The KMO is 0.848 and the Sig value of the Bartlett sphericity test is less than 0.001, indicating that the data are suitable for factor analysis. The maximum variance method is used to rotate and extract factors with eigenvalues greater than 1, and indexes with factor loadings not less than 0.5 as selection

criteria to select those with the strongest influence. Finally, seven indexes are obtained and then divided into two common factors, compliance-oriented and promotion-oriented environmentally responsible behavior, with variance contribution rates of 38.171% and 32.047%, respectively, and a cumulative contribution rate of 70.219%. The factor loading of each item after rotation is between 0.753–0.840, and the factor loading is greater than 0.500, indicating a suitability for factor analysis. The two dimensions of rural residents' environmentally responsible behaviors include seven items, which can be grouped as "compliance-oriented environmentally responsible behavior" and "promotion-oriented environmentally responsible behavior". For these items, the mean value of compliance dimension is 4.34 and the standard deviation is 0.618. The mean value of promoting environmentally responsible behavior dimension is 4.14 and the standard deviation is 0.743. By comparing the two, compliance-oriented environmentally responsible behavior has higher scores, better overall evaluation of residents, and smaller standard deviation, indicating that the residents' overall evaluation of this dimension is relatively consistent.

**Table 4.** Analysis of the overall characteristics of rural residents' environmentally responsible behavior.

| Environmentally Responsible Behavior | Mean | Standard Deviation | Factor Loadings | Variance Contribution Rate |
|---|---|---|---|---|
| Compliance environmentally responsible behavior | 4.34 | 0.618 | | 38.171% |
| E1 I have never littered plastic bags, cigarette butts, cigarette boxes, and other garbage in life | 4.38 | 0.735 | 0.840 | |
| E2 I will collect garbage and go to the garbage exchange supermarket for exchange | 4.34 | 0.692 | 0.827 | |
| E3 I collect and recycle waste paper | 4.29 | 0.732 | 0.812 | |
| Promoting environmentally responsible behavior | 4.14 | 0.743 | | 32.047% |
| E4 I will participate in volunteering activities related to garbage cleaning in the village | 4.21 | 0.868 | 0.753 | |
| E5 I will learn about garbage disposal and sorting | 4.07 | 0.905 | 0.809 | |
| E6 I will remind my friends to not litter | 4.16 | 0.887 | 0.775 | |
| E7 I will report the environmental issues and opinions in my area to the relevant departments | 4.12 | 0.960 | 0.825 | |

### 4.3. Confirmatory Factor Analysis

This study carries out confirmatory factor analysis on the relationship between the latent variables and their measurements. Before the factor analysis, the Cronbach's α value and combined reliability of each variable are first tested for reliability of the scale. The results show that the Cronbach's α value is 0.802, which was greater than 0.7, indicating that the measurement items have good correlation. The Bartlett sphericity test also shows a significant value of 0.000, which is less than 0.05, indicating that the data have good correlation and are suitable for factor analysis. Second, the indicators of each dimension are analyzed by SPSS and the Cronbach's α value of each dimension variable is obtained. From the perspective of validity, this study has a certain guarantee. From the perspective of content validity, a certain guarantee is obtained with the help of maturity scale, expert opinions, and group discussion, plus the verification of the results of the pre-survey. Finally, confirmatory factor analysis is carried out with Amos software to obtain the standardized factor loadings of each measurement variable. The combined reliability (CR) and average variance extraction value (AVE) of each dimension are calculated using the formula and Excel tool, where a CR value greater than 0.7 indicates high consistency, and an AVE value greater than 0.5 indicates that each dimension has discriminant validity.

In this study, perception is a variable with a multi-dimensional structure. Eight two-dimensional (2D) indicators were selected to measure the perception dimension. Given that this dimension is 2D, each first level needs to be evaluated, and the two sub-levels can be jointly validated for factor analysis. Table 5 shows the results. The Cronbach's α values of benefit and cost perceptions are 0.810 and 0.861, respectively. The CR values are 0.8096 and 0.8634, respectively, and each variable is greater than 0.7, indicating that the scale

has a good internal consistency. Thus, the model has good reliability. The standardized loading values of each variable range from 0.52 to 0.85, which are all greater than 0.5, indicating that each variable can explain the perception variable. Moreover, the AVE values of the perceptual variables are 0.4659 and 0.6783, respectively, which are greater than 0.5 and proves that this dimension has good validity.

**Table 5.** Inner structure fit index of rural residents' perceived attitudes and behavior measurement model.

| Latent Variable | Observed Variable | Factor Loadings | Cronbach'α | Combined Reliability (CR) | AVE Value |
|---|---|---|---|---|---|
| Benefit perception | B1 Spam exchange improves local image | 0.585 | 0.810 | 0.8096 | 0.4659 |
| | B2 Enhance residents' awareness of environmental protection | 0.520 | | | |
| | B3 After the supermarket was established, the garbage in the village was reduced | 0.710 | | | |
| | B4 Establishment of the supermarket contributes to the air protection | 0.755 | | | |
| | B5 Establishment of the supermarket contributes to the water protection | 0.808 | | | |
| Cost perception | C1 It takes time for learning and training related to garbage classification | 0.815 | 0.861 | 0.8634 | 0.6783 |
| | C2 It takes time to sort waste | 0.796 | | | |
| | C3 Garbage exchange needs to wait in line | 0.850 | | | |
| Satisfaction | A1 The exchange items in the garbage exchange supermarket are very practical and there are many types | 0.767 | 0.831 | 0.8318 | 0.5535 |
| | A2 The garbage exchange in the supermarket is very comprehensive | 0.686 | | | |
| | A3 The garbage exchange supermarket is very convenient and affordable | 0.787 | | | |
| | A4 The publicity model and management of the garbage exchange supermarket are very suitable | 0.731 | | | |
| Manner | D1 The establishment of garbage exchange supermarket contributes to the improvement of the environment in the region | 0.540 | 0.791 | 0.7889 | 0.4963 |
| | D2 I am satisfied with the current management of the garbage exchange supermarket by the supermarket manager | 0.520 | | | |
| | D3 I hope the garbage exchange supermarket will continue to run | 0.794 | | | |
| | D4 I support the establishment of a garbage exchange supermarket | 0.891 | | | |
| Compliance ERB | E1 I have never littered plastic bags, cigarette butts, cigarette boxes, and other garbage in my life | 0.880 | 0.791 | 0.8263 | 0.6154 |
| | E2 I will collect the garbage and go to the garbage exchange supermarket to exchange it | 0.703 | | | |
| | E3 I collect and recycle waste paper | 0.760 | | | |
| | E4 I will participate in volunteering activities related to garbage cleaning in the village | 0.670 | | | |
| Facilitated ERB | E5 I will learn about waste disposal and sorting | 0.796 | 0.838 | 0.8391 | 0.5665 |
| | E6 I will remind my friends to not litter | 0.733 | | | |
| | E7 I will report environmental issues and opinions in my area to the relevant departments | 0.778 | | | |

In this study, satisfaction is a one-dimensional structural variable. This study selects five measurement indicators to examine this dimension and builds a corresponding measurement model, as shown in Figure 3. Analysis using Amos and SPSS software shows that the Cronbach's α of satisfaction is 0.831 and the CR is 0.8318, both of which are greater than 0.7, indicating that the reliability of this dimension is high, and the model reliability is good. The standardized loading values of the four measurement variables range from 0.69 to 0.79, reaching a significant level, indicating that each measurement item can explain the satisfaction variable. Moreover, the AVE of this variable is 0.5534, which is greater than 0.5, indicating that this dimension has good validity.

In this study, attitude is a one-dimensional variable, and is explained using four variables. Factor analysis is carried out using SPSS, and an attitude model is constructed

by using Amos. The results show that the Cronbach's α value of attitude is 0.791, which is greater than 0.7, indicating that this dimension has a relatively high value and good consistency. Then, exploratory factor analysis is carried out and shows that the standardized loading value of each factor range from 0.52 to 0.89, which is greater than 0.5, and the CR value is 0.7889; thus, the measurement has good validity. The AVE value is 0.4963, which is close to 0.5, and indicates acceptable reliability.

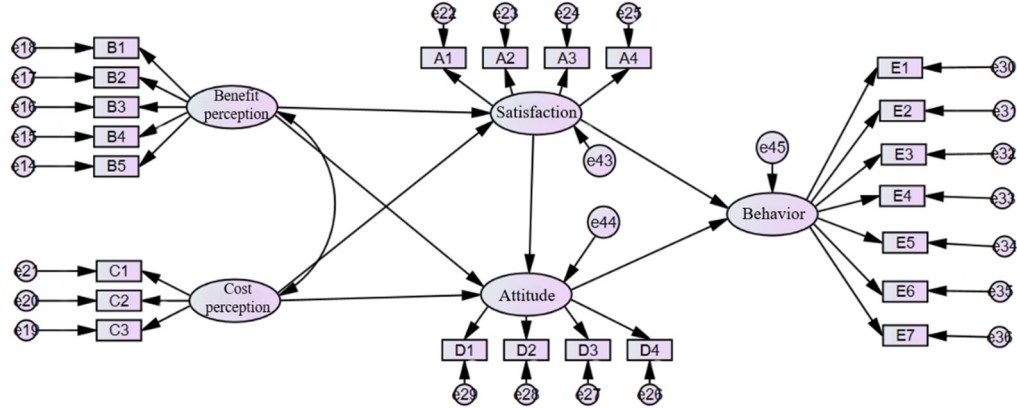

**Figure 3.** Model diagram of residents' perceptions and environmentally responsible behavior.

The dimension of environmentally responsible behavior is two-dimensional and two factors are extracted and tested separately. The overall Cronbach's alpha value of environmentally responsible behavior is 0.854, which is greater than 0.7, indicating that the scale has good internal consistency. The factor loadings of each variable range are 0.670–0.880, which are greater than 0.500, indicating that the scale has good internal consistency and validity. Next, the standardized loadings of the two dimensions of the compliance-type and the promotion-type environmentally responsible behaviors are calculated. The AVE values are 0.6154 and 0.5665, which are greater than 0.500, and the CR values are 0.8263 and 0.8391, which are higher than 0.7. Thus, the dimension of environmentally responsible behavior has good reliability and validity.

### 4.4. Structural Model Checking and Correction

This study has five latent variables in the measurement model, including the exogenous latent variables of cost and benefit perceptions and the endogenous latent variables of satisfaction, attitude, and environmentally responsible behavior. The measurement items are five for benefit perception, three for cost perception, five for satisfaction, four for attitude, and seven for environmentally responsible behavior. Figure 3 shows the structural equation model diagram.

The initial measurement model is calculated by using Amos, and Figure 4 shows the model fitting result. The fitness of the overall model shows that the chi-square value is 2.668, which is less than 3, and RMSEA = 0.072, which is less than 0.08; thus, good fitness is reached. However, in the initial model results, GFI = 0.850, AGFI = 0.814, CFI = 0.880, NNFI = 0.863, and IFI = 0.881, all are less than 0.9 and none reach the recommended value standard. Thus, the initial model needs to be adjusted and optimized. The initial model is corrected with reference to the correction index to further improve the overall accuracy. The addition of covariation relationships between e30 and e32, e28 and e29, and e30 and e31 shows the gradual improvement for each index. The modified model operation results show (see Table 6) that except for the values of AGFI and NFI which are lower than 0.9, the rest of the values are within the range of valid values, and the model fit is generally good and acceptable.

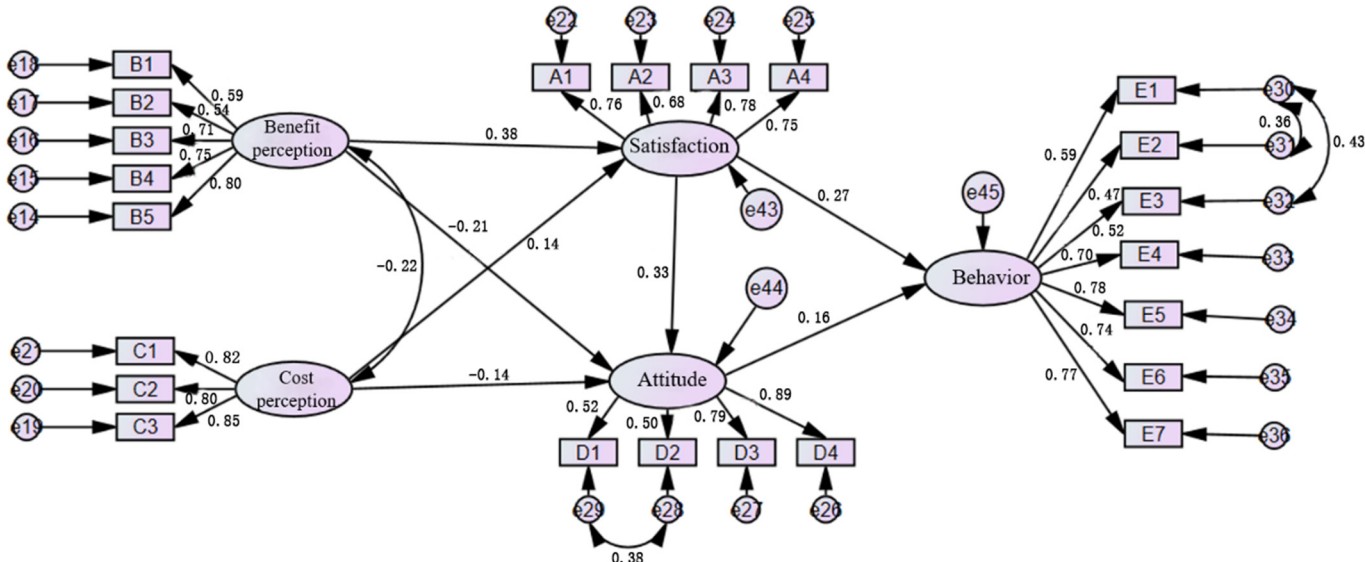

**Figure 4.** Residents' perceptions and environmentally responsible behavior modification model diagram.

**Table 6.** Reliability and validity analysis results of residents' perceptions and environmentally responsible behavior.

| Fit Metrics | CMIN/DF | GFI | AGFI | NFI | IFI | TLI | CFI | RMSEA |
|---|---|---|---|---|---|---|---|---|
| Judgment standard | <3 | >0.9 | >0.9 | >0.9 | >0.9 | >0.9 | >0.9 | <0.08 |
| Initial model | 2.668 | 0.850 | 0.814 | 0.822 | 0.881 | 0.863 | 0.880 | 0.072 |
| Corrected model | 1.891 | 0.937 | 0.874 | 0.876 | 0.937 | 0.927 | 0.937 | 0.053 |

*4.5. Structural Equation Model Analysis*

The model path analysis is further carried out on the modified model. Table 7 shows the results.

The results of the revised model show that residents' perceptions of the benefits of garbage exchange supermarkets have a positive effect on attitudes ($p < 0.05$), and the path critical value (CR) is 2.00, that is, H1 is supported. Residents participate in garbage exchange and perceive the benefits they obtain. Through analysis, rural residents believe that participating in garbage exchange can obtain certain social benefits (rural image) and environmental benefits (water and air improvement). In the benefit perception, residents have the deepest understanding of environmental benefit, and the explanatory value of reduction of waste in village is the largest. When asked about their attitude towards the garbage exchange supermarket during the field investigation, the residents said, "Since the establishment of the supermarket, basically no garbage can be seen in the village, and the environment has improved a lot. We think this supermarket is very well built, which is for the consideration of our people." At the same time, the relationship between the residents' perceptions of the benefits of garbage exchange supermarkets and residents' satisfaction is proven, $p < 0.001$, the path critical value (CR) is 5.660, and thus H4 is established. In terms of satisfaction with the services provided by the supermarket, the residents said, "We are very satisfied, the staff are all locals, and our daily garbage is directly exchanged on the exchange day, and the salt, soap and dishwashing liquid exchanged back are very useful, so there is basically no need to spend money to buy daily necessities".

Resident satisfaction and attitude towards garbage exchange supermarkets are also confirmed. The path coefficient is 4.439, $p < 0.001$, reaching saturation and significance, indicating that residents' satisfaction with garbage exchange supermarkets significantly affects residents' attitudes. Thus, H6 is established. During the interview, residents also

stated that the service personnel of the supermarket are all locals and the exchange items provided are household goods. The weekly exchange time is highly appropriate and the exchange places are in the central areas, which is very convenient. In general, residents have a positive attitude towards supermarkets and are also satisfied with the services, exchanges, and management models provided. This satisfaction can promote the residents' support for the garbage exchange supermarket to a certain extent, which also reflects the residents' attitudes towards this initiative.

**Table 7.** Path analysis results of structural model.

| Assumption | Path | Standardized Coefficient | CR Value | *p* Value | Result |
|:---:|:---:|:---:|:---:|:---:|:---:|
| H1 | Benefit perception → Attitude | 0.090 | 2.000 | * | Yes |
| H2 | Cost perception → Attitude | 0.041 | −2.153 | 0.312 | No |
| H3 | Attitude → Environmentally responsible behavior | 0.047 | 2.240 | * | Yes |
| H4 | Benefit perception → Satisfaction | 0.077 | 5.660 | *** | Yes |
| H5 | Cost perception → Satisfaction | 0.036 | −3.286 | ** | Yes |
| H6 | Satisfaction → Attitude | 0.083 | 4.439 | *** | Yes |
| H7 | Satisfaction → Environmentally responsible behavior | 0.056 | 3.3598 | *** | Yes |

Note: * means $p < 0.05$, ** means $p < 0.01$, *** means $p < 0.001$.

Second, the relationship between residents' cost perceptions and attitudes toward garbage exchange supermarkets is not verified. The path critical ratio (CR) value of cost perception to support attitude is −2.153 and the *p* value is 0.312, which is greater than 0.05. Therefore, residents' perceptions of the cost of participating in garbage exchange is unrelated to residents' attitudes; therefore, H2 does not hold. Cost and benefit perceptions are the manifestations of different perception dimensions. According to social exchange theory, when residents engage in an activity, the trade-off between perceived costs and perceived benefits determines their participation. In the context of garbage exchange supermarkets, residents believe that the cost of participating in the exchange can be ignored. At the same time, the on-site investigation reveals that, due to the inconsistent distance from the exchange point, residents who are relatively close to the supermarket generally do not need to queue and wait for those who are farther away to complete the exchange; however, the residents who are a little further away go earlier and queue in front of the supermarket. In this dimension, residents' perceptions vary greatly, but overall, the distance, and thus the need to queue, affects the residents' exchange frequency to a certain extent, but not their attitudes. Residents who live farther away also have a positive attitude towards supermarkets. However, there is a correlation between residents' perceptions of the cost of garbage exchange supermarkets and residents' satisfaction. The analysis showed that the *p*-value between the two was 0.001, the $p < 0.01$, and the path critical ratio (C.R.) value was −3.286, indicating that cost perception was negatively correlated with satisfaction, assuming that H5 was confirmed. This may be due to the occasional need for queuing, the corresponding time required for garbage classification, and the distance from the exchange point affects the completion time of the transaction, which can affect the residents' exchange experience and thus affect their satisfaction. Regarding the uneven distribution of supermarkets, which causes inconvenience for several residents, the organizers from Xixinan village pointed out that the garbage exchange supermarket is currently in a stage of rapid development. In the future, the initiative will be extended to every village as far as possible to meet the exchange needs of residents in different places and reduce the exchange cost.

Finally, the results of the path analysis show that the standardized coefficient of residents' attitudes towards garbage exchange supermarkets and the path of environmentally responsible behavior is 2.240 and the *p* value is 0.0242, which is less than 0.05. Therefore, the positive effect of residents' attitudes towards garbage exchange supermarket on residents' environmentally responsible behaviors is confirmed. Thus, H3 is established. Findings also prove the relationship between residents' satisfaction with garbage exchange supermarkets and their environmentally responsible behavior, with the path coefficient of 3.3598 and the

*p*-value less than 0.001, indicating that satisfaction has a positive effect; therefore, H7 is confirmed. Furthermore, the relationship between attitude and behavior has been widely confirmed in different fields. In this study, D3 (I hope the supermarket can continue to run) and D4 (I support the establishment of a garbage exchange supermarket) can directly affect whether residents participate in the exchange. For the dimension of satisfaction, the management model of garbage exchange supermarkets and the types of exchange items can also affect the behavior of residents. Moreover, examining the behavior of residents in this study not only includes whether to go to the exchange, but also measures the derived environmental behavior, such as whether to persuade friends to participate in volunteering activities. Regarding the behavior of environmental responsibility, the residents in the field survey indicate that they are willing to collect garbage and exchange, which also shows their strong awareness of the need for environmental protection. Therefore, the establishment of the garbage exchange supermarket effectively influences residents to be environmentally responsible.

## 5. Conclusions and Discussion

### 5.1. Conclusions

Using a garbage exchange supermarket as an example, this study reveals residents' perceptions of garbage exchange supermarkets and their environmentally responsible behaviors by means of quantitative statistical analysis methods. Using Amos software, a relational model is constructed with perception, satisfaction, attitude, and environmentally responsible behavior variables. The following conclusions are drawn.

(1) The dimension of residents' perceptions is mainly divided into two aspects: benefit and cost perception. Residents' perceptions of the benefits of the garbage exchange supermarkets are strong and positive, in which residents' perceptions of environmental benefits are stronger; their awareness of environmental protection is strong; and their perceptions of the costs of the garbage exchange supermarkets are weak. The attitude towards the garbage exchange supermarket is highly positive and the satisfaction is high. Residents are quite satisfied with the exchange items, publicity mode, management model, exchange items, and types of convertible wastes in the garbage exchange supermarket. Overall, residents are willing to pay the cost to go to the supermarket to exchange. In addition, residents' environmentally responsible behaviors are divided into compliance-type and promotion-type environmentally responsible behaviors, among which the former shows higher willingness and internal consistency.

(2) Constructing and verifying the model of residents' perceptions and behaviors show that:
   ① Residents' perceptions of the benefits of garbage exchange supermarkets positively affect their attitudes and satisfaction. Cost perception is unrelated to attitude but affects their satisfaction.
   ② Residents' attitudes towards garbage exchange supermarkets positively affect their environmentally responsible behaviors.
   ③ Residents' satisfaction with garbage exchange supermarkets affects their attitudes and positively affects their environmentally responsible behaviors.

### 5.2. Discussion

Considering garbage exchange supermarkets as an example, this study explores the new garbage disposal models emerging in rural areas in China. The results show that the garbage exchange supermarket relies on material incentives to encourage residents to participate in waste management behavior, which can produce a certain spillover effect and improve the environmentally responsible behavior of villagers. The garbage exchange supermarket not only plays an important role in waste management in the area, but also allows residents to change from passive environmental protection to active environmental protection and enhances residents' awareness of environmental protection and improves the image of the area. In China, this practice is not limited to Huangshan in Anhui. It originated from the exploration of rural grassroots movements in Fengxian in Shaanxi and

is now widely popular in rural areas such as Zhejiang Province, Jiangsu Province, and Guizhou Province. All the initiatives have contributed to an improvement in the local rural living environment. However, compared with rural garbage exchange supermarkets in other places, Huangshan City's approach is more distinctive, more secure, and sustainable, and has been approved by the Secretary of the Anhui Provincial Party Committee and promoted in rural areas of the province. This is due to the launch of the first cross-provincial river basin ecological compensation system pilot project in China, the Xin'an River-Qiandao Lake Ecological Compensation Pilot Zone. The government played a significant leading and organizational role, reducing the source of rural pollution by encouraging garbage exchange supermarkets, improving the villagers' environmental awareness and knowledge of garbage classification and promoting ecological protection and environmental construction in the upper reaches of the Xin'an River. At the same time, the Huangshan Municipal Government handed over the garbage disposal work to Zhonghuanjie Group, giving full play to the synergy between the market and the government. Zhonghuanjie Group carries out safe waste treatment, ensuring the effective disposal of exchanged garbage and effectively form the government's new "Public–Private Partnerships" model. The rural garbage exchange supermarket, the unified management of pesticides in rural areas, and the relocation of alpine farmers together constitute an important part of the "Xin'anjiang Model" of cross-basin ecological protection, which is a good model for the living environment and ecological protection of the river basin in other rural areas.

This study also found several problems with garbage exchange supermarkets. For example, most residents are dissatisfied with the ratio of waste exchange; supermarkets have limited operating funds and cannot fully meet the needs of residents for exchange. Moreover, to participate in the "Swap for the sake of exchange", many residents ask merchants for more plastic bags when buying other items. The use of plastic bags in this area has thus increased. Residents also demand more types of items that can be exchanged for garbage. In addition to daily garbage, how to extend this initiative to rural environmental management such as construction waste, wastewater, and sewage needs to be determined. These problems need to be seriously considered by government managers and relevant supermarket operators. Inappropriate policies result in negative externalities (such as the increased use of plastic bags, etc.), which will affect the goals and original intentions of rural garbage exchange supermarkets. Some measures could further improve the efficiency of garbage exchange supermarkets and provide better services for residents.

(1) First of all, it is necessary to increase the publicity regarding the need for environmental protection to the villagers and emphasize the importance of rural garbage exchange supermarkets in improving the living environment in rural areas. Let people realize that running a supermarket is a means to protect the environment and a provide a beautiful home, not an end in itself, and guide villagers to realize the dangers of plastic bags, which cannot be swapped for the sake of exchange, and reduce the generation of negative externalities.

(2) The activities related to garbage exchange supermarkets (such as agricultural product exhibition, lectures on garbage classification) can be increased. The survey shows that in addition to garbage exchange, residents of Xixinan village in Huizhou District also participate in activities such as agricultural product exhibition and sale, which increases the residents' sense of participation. However, the garbage exchange supermarkets in Liukou and Huangtian villages have a single function, i.e., garbage exchange.

(3) For other non-convertible wastes in rural areas (such as construction waste), targeted collection and transportation must be carried out.

(4) The types of exchange items available in supermarkets can be expanded to consider the needs of different age groups. Moreover, the capital source of the supermarket is the key to its sustainable operation, which needs more consideration and multi-channel financing.

It is reported that in 2016, the annual volume of domestic waste in rural China was 150 million tons, half of which could not be treated, and the volume of domestic waste in most rural areas in China is still in the rising stage of an inverted "U" curve [53]. Therefore, for countries all over the world, including China, the issue of improving the rural living environment is a long-term effort and there is still a long way to go in the development of the garbage exchange supermarket model.

The perceptions, attitudes, and behaviors of rural residents are closely related. It is necessary to cultivate villagers' environmental awareness to form a positive attitude and to transform passive environmental protection into active environmental protection, as well as to promote low-carbon consumption and green development in rural areas.

**Author Contributions:** S.L. provided the research idea, outlined the whole research paper, and solved technical problems. Z.Z. contributed to the field research and the data collection and wrote the first draft of this paper. Y.L. proofread the entire text, including a check of data and literature sources. All authors have read and agreed to the published version of the manuscript.

**Funding:** This research study was supported by the National Natural Science Foundation of China (Grant no. 41971171).

**Institutional Review Board Statement:** Not applicable.

**Informed Consent Statement:** Participants were informed that participation in the study was voluntary and that completing the questionnaire face-to-face indicated their informed consent to participate this study.

**Data Availability Statement:** The data presented are available on request from the corresponding author.

**Conflicts of Interest:** The authors declare no conflict of interest.

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
