# Peer review of "Rural Residents’ Perceptions, Attitudes, and Environmentally Responsible Behaviors towards Garbage Exchange Supermarkets: An Example from Huangshan City in China"

_sustainability, doi:10.3390/su14148577_

Round 1
Reviewer 1 Report
The article is interesting, scientifically processed with a statistically evaluated survey. I also evaluate it positively model diagram of residents’ perception and environmental responsibility behavior. This study deals with 7 hypotheses. The formulated hypotheses are justified due to the topic of the article. The article is an inspiration for the development of waste management at the municipal level.
Reviewer 3 Report
Review report
Manuscript title: Rural Residents’ Perceptions, Attitudes, and Environmental Responsibility Behaviors towards Garbage Exchange Supermarkets: An Example from Huangshan City in China
Major comments:
· The abstract does not include any information on the methodology used; it must explain the questionnaire survey and random sampling method used and the total number of valid participants questionnaire collected.
· P6: Study Design: must be shortened or moved to the introduction, especially L244 to L282.
· P1, L22: The author claims that “ This study can provide useful reference n the conclusion….beautiful rural construction” how does this conclusion been extracted? Which question? Any statistical significance with the cost?
· P12 and P13 Table3 and Table 4. Remove Minimum and Maximum since those are the answer scale of 1-5
· Some of the questions used are enormously not reliable, such as “B4 After the establishment of the supermarket, the air is cleaner” how did the author link the collection of garbage will reduce air pollution? Did he share any air pollution data before and after with the participants? Need to justify how those B4 and B5 questions were selected?
· Of the study participants, 74.1% are female; how can the author justify that this study is random???
· Reference numbers 1, 2, and 3 are irrelevant to the current study and need to be removed from the introduction part.
Minor comments:
P1L3: Title: correct: “ Supermarkets1” to: “Supermarkets”
P1, L12: Change:” perception for the” to: “perception of the”.
P1, L22: Change:” provide useful” to: “provide a useful”.
P2, L55: Repace: Thus, the garbage is collected effectively and is classified at the same time while the villagers not only gain benefits but also improve their rural living environment. With: “Thus, the garbage is collected effectively and classified simultaneously while the villagers not only gain benefits but also improve their rural living environment”.
P2, L79: Change: “perspective of residents”, to: “residents' perspective”
P4, L179: change: “not only an economic” to: “not only an economic”
P5, L204: change: “has a direct effect on” to: “directly affects”.
P5, L266: change: “found that that” to: “found that”
P5, L229: change: “Satisfaction not only affects such participation in garbage exchange” to: “Satisfaction affects not only such participation”
P6, L251: Change: “and is known as the national forest city” to: “known as the national forest city”
P6, L263” change: “main stream” to: “mainstream.”
P9, L368: change: “Exclusion of those” to: “ The exclusion of”
P11, L406: Rewrite the sentence to avoid a dangling modifier: “To measure the reliability of the scale and each dimension index, SPSS software was used to test the items, and the overall reliability was analyzed”
Round 2
Reviewer 3 Report
The current version of the manuscript is well improved.